# Genome-Wide Identification of the *PYL* Gene Family in *Chenopodium quinoa*: From Genes to Protein 3D Structure Analysis

**Gastón Alfredo Pizzio** [1,2]

1 Instituto de Biología Molecular y Celular de Plantas, Universidad Politécnica de Valencia (UPV-CSIC), CP46022 Valencia, Spain; gapizzio@gmail.com or gaspiz@ibmcp.upv.es
2 CIT-RIO NEGRO Sede Atlántica, Universidad Nacional de Río Negro (UNRN-CONICET), Viedma CP8500, Río Negro, Argentina

**Abstract:** The halophytic crop *Chenopodium quinoa* has a remarkable resistance to harsh growth conditions in suboptimal environments and marginal soils. Additionally, quinoa is a pseudocereal and produces seeds with outstanding nutritional value. Quinoa is an allotetraploid (2n = 4× = 36) with an estimated genome size of approximately 1.5 Gbp. In plants, the family of pyrabactin resistance 1 (PYR1)/PYR1-like (PYL)/regulatory components of ABA receptors (RCAR) play a vital role in the initial step of ABA signaling, leading to abiotic stress resistance. Here 20 *CqPYL* genes were identified using the genome-search method. Based on the phylogenetic analysis, these *CqPYL* genes were divided into three classes or subfamilies. These genes have different structures and intron numbers, even within the same subfamily. Analysis of conserved motifs showed the presence of the PYR_PYL_RCAR motif domain in each PYL protein sequence. Furthermore, the tissue-specific expression of CqPYLs was analyzed through public available RNA-seq data. CqPYL4a/b and CqPYL8c/d showed higher expression levels in seedlings. Finally, 3D structures of the CqPYL proteins were predicted by homology modeling and analyzed through topology inspection to speculate on putative new ABA receptor features. This study provides a theoretical basis for further functional study of PYL genes for stress-resistance breeding of quinoa and other crops.

**Keywords:** *Quinoa*; *PYR/PYLs* gene family; abiotic stresses; abscisic acid perception; bioinformatics





## 1. Introduction

In the last years, quinoa (*Chenopodium quinoa Willd.*) demand and production have risen notably around the world [1]. Quinoa is an annual crop of great value in regard to tolerance to abiotic stresses and adaptability to unfavorable environments. It is resilient to harsh growth conditions in terms of soils, rainfall, temperature, and altitude. It can thus grow from sea level up to 4500 m above sea level and can adapt to frost, drought, and salinity [2]. For instance, quinoa achieves the highest biomass when grown in the presence of 100 mM NaCl and shows only 20~50% biomass reduction even when treated with 500 mM NaCl (sea-water salt concentration) [3]. Moreover, quinoa is one of the most nutritious food crops currently known [4]. Its high-quality protein content makes it a good crop for enhancing high-quality plant protein food production [5]. Moreover, quinoa seeds also provide all essential amino acids in contrast to cereals [6]. Quinoa lacks gluten; hence it is suitable for gluten-free (GF) diets for people with celiac disease, wheat allergies, or gluten ataxia [7]. Even more, as a whole grain food, quinoa contributes to overcoming some nutritional inadequacies of GF products [8]. It is indeed a source of high levels of dietary fiber [9], vitamins, minerals, and antioxidants [10].

The phytohormone abscisic acid (ABA) is involved in plant stress response, growth and development. Among the plant physiological processes, ABA plays a role in seed

germination and early seedling growth, stomata closure, shoot and root growth and development, senescence, fruit ripening, fruit and leaf abscission, and bud dormancy. Moreover, ABA is a key player in the induction of drought tolerance and other biotic and abiotic stresses [11–13]. In the ABA core signaling pathway, ABA receptors are the key component, perceiving the hormone and triggering the signaling cascade and response. ABA perception is carried out through soluble receptors called pyrabactin resistance 1 (PYR1)/PYR1-like (PYL)/regulatory components of ABA receptors (RCAR) [14–16]. Upon ABA perception, a ternary complex is formed between ABA-PYL and clade A protein phosphatase type 2Cs (PP2Cs). The formation of this ternary complex relieves the inhibition of ABA-activated subclass III SNF1-related protein kinases 2 (SnRK2s) by PP2CAs [17,18]. Then, activated SnRK2s induces the activation of a battery of ABA effectors from transmembrane channels to transcription factors [12]. In turn, the activity and half-life of components of this ABA core signaling pathway are regulated by multiple protein kinases, the ubiquitin-26S proteasome system, the endocytic/vacuolar degradation pathway, and the circadian system [12,19–21].

ABA signaling plays a key role in plant response to abiotic stress. On the other hand, plant agriculture and food production are being challenged by climate change. Thus, studies on crops well adapted to harsh environments, such as the halophytic pseudocereal quinoa, are needed to further characterize the role of ABA receptors to prospect new features in plant adaptation to environmental stress. Thereby, this study aims to unveil the complex gene family of ABA receptors in *Chenopodium quinoa*. The PYL family has been systematically identified in many crops, including 14 PYLs in tomato [22], 12 PYLs in rice [23], 29 PYLs in tobacco [24], 46 PYLs in canola [25], 8 PYLs in grape [26], 38 PYLs in wheat [27] and 23 PYLs in *N. benthamiana* [28]. However, the PYL gene family in quinoa has not been studied up to day, hampering its application in quinoa stress improvement. Quinoa is an allotetraploid (2n = 4× = 36) with an estimated genome size of approximately 1.5 Gbp. The completion of the reference genome sequencing of quinoa lays the foundation for analyzing the genomic organization of the PYL family in this crop at the genomic level [29,30].

In this study, it was identified 20 PYLs in quinoa, using the genome-search method based on the available latest genome information in ChenopodiumDB (kaust.edu.sa). Phylogenetic and gene structure analyses were performed to classify this CqPYL in the three classical subfamilies of ABA receptors. Then, conserved domain analysis and visual inspection of protein sequence alignment were carried out to assess the potential of each CqPYLs to be functional. Furthermore, the tissue-specific expression patterns of CqPYLs were systematically investigated based on public RNA-seq samples. Finally, we used the homology modeling method to predict the protein structure of the CqPYLs and perform a topology analysis. This study provides the fundamental basis for the further functional study of CqPYLs, and also for stress tolerance improvement of quinoa.

## 2. Materials and Methods

### 2.1. Genome-Wide Identification of PYL Genes in Chenopodium Quinoa

The *Arabidopsis thaliana* and *Nicotiana benthamiana* PYL protein sequences (described in [28]) were used as the queries to perform a BLASTP search against the local protein database of *chenopodium quinoa* [30]: AtPYR1 (AT4G17870), AtPYL1 (AT5G46790), AtPYL2 (AT2G26040), AtPYL3 (AT1G73000), AtPYL4 (AT2G38310), AtPYL5 (AT5G05440), AtPYL6 (AT2G40330), AtPYL7 (AT4G01026), AtPYL8 (AT5G53160), AtPYL9 (AT1G01360), AtPYL10 (AT4G27920), AtPYL11 (AT5G45860), AtPYL12 (AT5G45870), AtPYL13 (AT4G18620), NbPYL1a (NbD001180), NbPYL1b (NbD012105), NbPYL1c (NbD050776), NbPYL1d (NbD020143), NbPYL2a (NbD047977), NbPYL2b (NbD011975), NbPYL2c (NbD053066), NbPYL2d (NbD003819), NbPYL4a (NbD015993), NbPYL4b (NbD014593), NbPYL4c (NbD006558), NbPYL4d (NbE03056248), NbPYL4e (NbD045610), NbPYL4f (NbD010048), NbPYL4g (NbD036501), NbPYL8a (NbD015117), NbPYL8b (NbD008843), NbPYL8c (NbD004872), NbPYL8d (NbD027042), NbPYL8e (NbE05066286), NbPYL8f

(NbD028217), NbPYL8g (NbD030589) and NbPYL8h (NbD040073). Protein sequences with an E value > 0.001 were chosen. Then, the protein sequences identified were integrated and parsed by manual editing to remove the redundancy. The 20 remaining proteins were considered as candidate quinoa PYL proteins. The molecular weight (Mw) and isoelectric point (pI) of these identified proteins were obtained using ExPASy online software [31].

### 2.2. Phylogenetic Relationship and Gene Structure of CqPYLs

The chromosome physical location of the 20 CqPYL was obtained from the quinoa genome DB annotation files and displayed in a handmade scheme. In order to determine the phylogenetic relationship of the CqPYLs, multiple sequence alignments of the identified 20 PYL proteins of quinoa and 14 PYL proteins of Arabidopsis were performed using ClustalW tool and the maximum-likelihood method (ML). Phylogenetic tree was constructed by MEGA X software (version 10.2.6, [32]) with a bootstrap value of 1000. Genetic distance matrix of CqPYLs CDS sequences analysis was conducted using MEGA X software with the Tamura–Nei model [33] and a bootstrap procedure of 1000 replicates. The Genetic distance heatmap plot was performed with the genetic distance data using CLUSTVIS web tool [34]. The paralogous gene pairs were identified based on the physical chromosome position and sequence similarity. The gene UTR–exon–intron structure of these 20 CqPYL genes was obtained from the quinoa genome DB annotation files and displayed using the online software Gene Structure Display Server (GSDS2.0) [35].

### 2.3. Conserved Motifs Analysis and Expression Profiles of PYL Genes in Quinoa

The MOTIF search tool was used to search in pfam and NCBI-CDD databases for the presence of PYR_PYL_RCAR motif domain in each PYL protein sequence. PYR_PYL-RCAR motif includes the Polyketide cyclase/dehydrase and lipid transport motif and the Bet_v1-like motif of major pollen allergen of *Betula verrucosa*.

To understand the tissue-specific expression patterns of these identified CqPYLs, available RNA-seq samples from NCBI Sequence Read Archive database of different tissues were analyzed: 15 h old young seedlings (SRR10493379; sterilized seeds were embedded on solid 1/2 MS medium for 15 h), 1 week old seedlings (SRR5974428, SRR5974429, and SRR5974434), seedlings root (SRR11050561), seedlings shoot (SRR11050558), stem and leaf from 6-week-old plants (SRR5974425 and SRR5974435, respectively). The number of total reads aligned to each CqPYL CDS sequence was extracted, and the RPKM (reads per kilo base of transcript per million fragments mapped reads) value was calculated and plotted. To visualize the tissue-specific CqPYLs expression a heat map was constructed using CLUSTVIS web tool [34].

### 2.4. Protein Alignments and Homology Modeling of CqPYLs 3D Structure

Protein alignments were performed with MegAlign 7.0 software, from the DNASTAR laser gene suite, by clustal W method. Graphics showing amino acid sequence alignment of AtPYLs and CqPYLs, and secondary structure alignment of ABA receptors were generated using the Easy Sequencing in PostScript program (ESPRIPT [36]). The secondary structure of the NbPYLs is predicted according to the crystallographic structure of AtPYR1 (Protein DataBank Code 3k3k), AtPYL5 (4jdl) or AtPYL9 (3x9r) as appropriate. In order to obtain 3D structure, CqPYL protein homology modeling was performed using Swiss-Model interactive tool [37]. The models of CqPYL1a, CqPYL4a and CqPYL13 were obtained using the topology of SlPYL1 (5mob) as template; for CqPYL8a and CqPYL5a, protein 3D structure template AtPYR1 (3k3k) and CsPYL1 (6zuc) were used, respectively. Quality of each protein structure model was estimated with a QMEANDisCo global score function. We also calculated QMEANDisCo local (per residue) values and some QMEAN z-scores: QMEAN, Cβeta, All Atom, Solvation, and Torsion. The QMEAN Z-score provides an estimate of the 'degree of nativeness' of the structural features observed in the model. Higher QMEAN Z-scores indicate better model structure [38,39]. Based on these scores, we can assume that all protein models generated were reliable. The 3D structure was

analyzed and displayed by Pymol software (PyMOL Molecular Graphics System, Version 1.6 Schrödinger, LLC[9]). Structural alignments were performed using the PyMOL's plugin PyMod 2.0 [40].

## 3. Results

### 3.1. Genome-Wide Identification of PYLs in Quinoa

A total of 20 PYL genes were identified in the quinoa genome, and the sequence characteristics of PYL open reading frames (ORFs) and proteins were investigated and compared (Table 1). The ORF and protein length of CqPYLs had some variations that ranged from 288 bp to 900 bp (average 561 bp) and from 95 to 299 aa (average 186 aa), respectively. Accordingly, the molecular weight (Mw) ranged from 10.79 to 32.06 kDa. The isoelectric point (pI) ranged from 4.98 to 7.03, suggesting no significant difference occurred among them. The chromosome localization showed that the identified 20 CqPYLs were unevenly distributed in 12 out of 18 quinoa chromosomes (Table 1 and Figure 1). Chromosome 1 (Ch1) possessed a high number of PYL genes containing six CqPYL genes; Ch4, 10, and 16 host two CqPYL each; Ch2, 3, 5, 7, 8, 13, 15, and 18 carry only one CqPYL each. On the other hand, there are no CqPYLs in Ch6, 9, 12, 14, and 17 (Table 1 and Figure 1).

**Table 1.** Basic information of CqPYLs.

| | | | | Deduced Protein | |
|---|---|---|---|---|---|
| **Gene ID** | **Gene Name** | **Genome Location** | **ORF Length (bp)** | **Size (aa)** | **pI/Mw (KDa)** |
| AUR62007289 | PYL1a | Chr13:2720366..2721031 (- strand) | 666 | 221 | 5.85/24.48 |
| AUR62018717 | PYL1b | Chr16:75866105..75874073 (+ strand) | 900 | 299 | 5.68/32.06 |
| AUR62022062 | PYL2a | Chr04:12623848..12625718 (- strand) | 405 | 134 | 4.72/14.84 |
| AUR62009160 | PYL2b | Chr01:108286563..108297915 (+ strand) | 744 | 247 | 7.03/27.30 |
| AUR62044383 | PYL2c | Chr01:100931182..100949105 (- strand) | 816 | 271 | 7.02/29.91 |
| AUR62023922 | PYL2d | Chr15:23036300..23036854 (- strand) | 555 | 184 | 6.31/20.63 |
| AUR62038318 | PYL2e | Chr18:7819431..7819985 (+ strand) | 555 | 184 | 5.95/20.62 |
| AUR62030521 | PYL4a | Chr16:62185514..62187006 (- strand) | 486 | 161 | 5.48/17.40 |
| AUR62030771 | PYL4b | Chr08:19769818..19771332 (- strand) | 483 | 160 | 5.92/17.30 |
| AUR62012310 | PYL5a | Chr03:78021315..78022708 (+ strand) | 651 | 216 | 5.25/23.49 |
| AUR62022950 | PYL5b | Chr04:2113036..2114588 (+ strand) | 645 | 214 | 5.24/23.25 |
| AUR62006143 | PYL8a | Chr07:72040444..72043736 (+ strand) | 561 | 186 | 5.75/20.91 |
| AUR62019345 | PYL8b | Chr05:79849351..79852678 (+ strand) | 561 | 186 | 5.75/20.91 |
| AUR62023479 | PYL8c | Chr10:2085777..2089495 (- strand) | 507 | 168 | 6.16/18.83 |
| AUR62004471 | PYL8d | Chr01:119133919..119137147 (- strand) | 510 | 169 | 5.97/18.93 |
| AUR62013633 | PYL8e | Chr10:12344051..12346110 (- strand) | 546 | 181 | 5.87/20.52 |
| AUR62004215 | PYL8f | Chr01:116546522..116547328 (+ strand) | 444 | 147 | 5.31/16.62 |
| AUR62027689 | PYL11 | Chr01:127197604..127197957 (- strand) | 354 | 117 | 4.98/13.20 |
| AUR62027690 | PYL12 | Chr01:127151375..127151662 (- strand) | 288 | 95 | 4.91/10.79 |
| AUR62031217 | PYL13 | Chr02:8102358..8102903 (+ strand) | 546 | 181 | 5.95/20.42 |

### 3.2. Phylogenetic and Gene Structure Analysis of CqPYLs

In order to explore the phylogenetic relationship of genes in quinoa, a maximum-likelihood (ML) tree was constructed using the full-length protein sequence alignments of the identified 20 CqPYLs and 14 AtPYLs (Figure 2). Since there is no standard nomenclature for PYLs in quinoa, these identified genes were named based on their phylogenetic relationship between AtPYLs and CqPYLs. Consistent with the previous studies in Arabidopsis

and other species [13–15,28,41], CqPYLs could be classified into three subfamilies. In detail, subfamily I contained six members (CqPYL8a-f), and subfamily II included eight members (CqPYL4a-b, CqPYL5a-b, CqPYL11, CqPYL12, and CqPYL13), and the remaining six ABA receptors belonged to subfamily III (CqPYL1a-b and CqPYL2a-e).

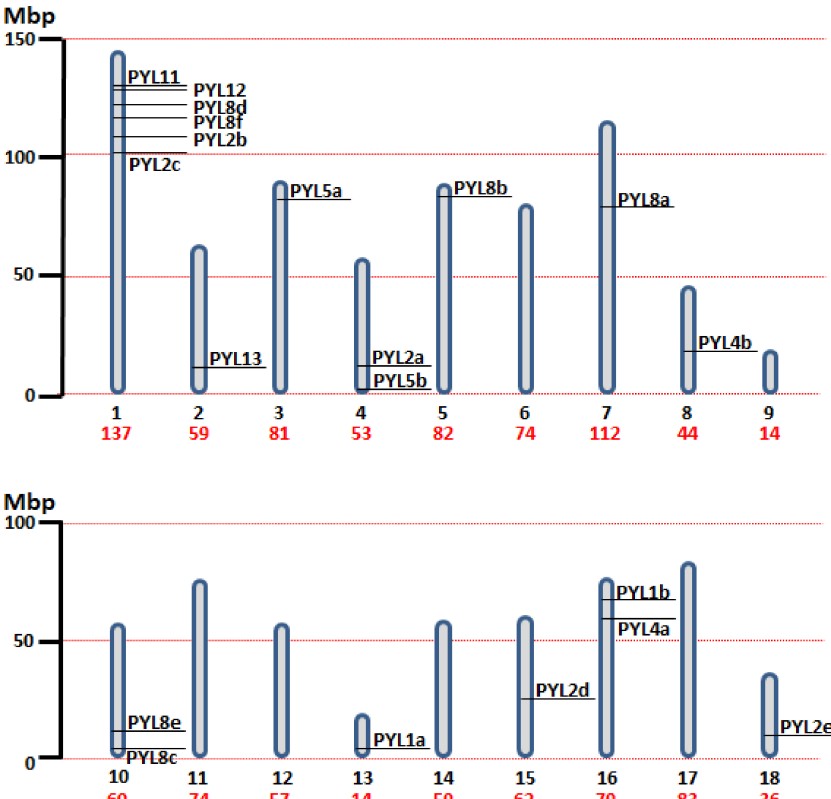

**Figure 1.** Chromosomal distribution of the identified 20 CqPYL genes across the quinoa genome. The 18 chromosome of quinoa are shown to scale based on their physical size. Mbp: Mega base pair. Under each chromosome in black the chromosome number and in red the chromosome size.

Quinoa resulted from the hybridization of the ancestral diploid genomes A and B [42]. Accordingly to the genomic in situ hybridization results, possibly from *C. nevadense or C. watsonii* as the donor of genome A and from *C. suecicum, C. ficifolium*, or *C. virideas* donor of genome B [42]. Using mutation rates, it was estimated the tetraploidization and the whole-genome duplication event to have occurred 3.3–6.3 million years ago [29]. In this work, nine and eleven CqPYL genes were identified from A and B subgenomes, respectively, suggesting some degree of variation occurred on the PYL gene abundance at the subgenome scale (Figure 3). Moreover, based on the sequence similarity and chromosomal position of CqPYLs, genomic segmental duplication or gene tandem duplication likely has occurred in quinoa. The analysis of PYL gene duplication showed that there were four duplications events, giving rise to paralogous genes. In subgenome A at chromosome 10 (Ch10) with CqPYL8c/e and in subgenome B at Ch1 with CqPYL11/12, CqPYL8d/f and CqPYL2b/c (Figures 1 and 3). On the other hand, based on the phylogenetic relationship of CqPYL CDS sequences, it was determined the homoeologous sub-groups of quinoa PYL genes (Figure 3 and Figure S1). A total of eight sub-groups were proposed, with each containing the A and B homoeologous CqPYLs copies (Figure 3). The homoeologous CqPYLs sub-groups are composed by: CqPYL1a-b, CqPYL2a-c, CqPYL2d-e, CqPYL4a-b, CqPYL5a-b, CqPYL8a-b, CqPYL8c-f and CqPYL11-13.

The UTR–exon–intron gene structure is an important evolutionary feature in revealing the relationship between members of a given gene family [43]. Thus, the UTR–exon–intron organization of the CqPYL family was further analyzed (Figure 4). Results showed that

there are six intronless genes (CqPYL1a, CqPYL2d-e, CqPYL11, C1PYL12, and CqPYL13), seven genes with only one intron (CqPYL1b, CqPYL2a, CqPYL4a-b, CqPYL5a-b, and CqPYL8f), four genes with two introns (CqPYL8a-b/d-e), two genes with three introns (CqPYL2b and CqPYL8c) and one gene with five introns (CqPYL2c).

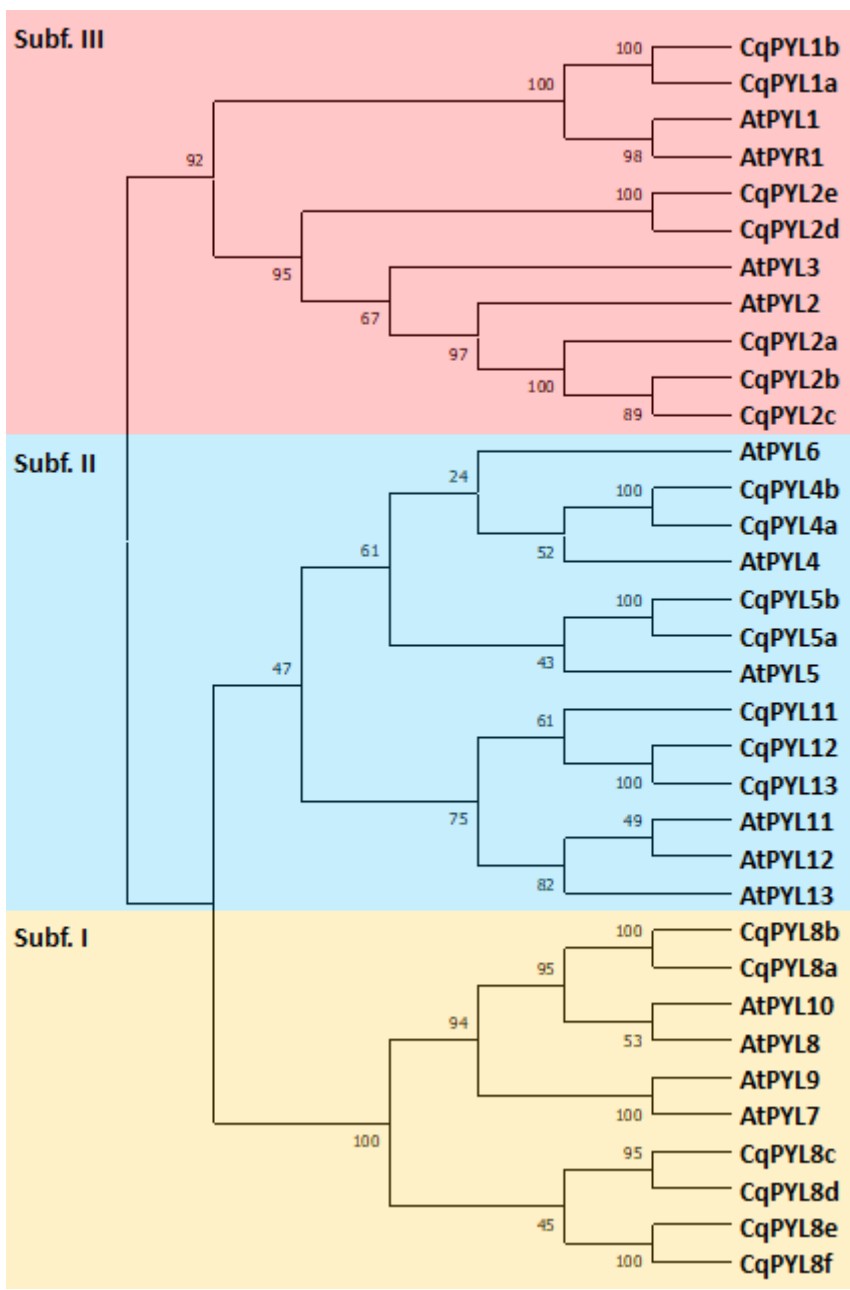

**Figure 2.** Phylogenetic analysis of PYL gene family in *Chenopodium quinoa* and *arabidopsis thaliana*. Tree was constructed by MEGAx using maximum-likelihood method with 1000 bootstraps. Orange, blue and red clades represent subfamilies I, II and III, respectively. The percentage of replicate trees in which the associated taxa clustered together in the bootstrap test (1000 replicates) are shown next to the branches.

Interestingly, genes in the same subfamily do not show similar gene structures. Additionally, intron length and exon–intron structures are highly conserved during evolution between the homoeologous gene CqPYL2d and e, CqPYL4a and b, CqPYL5a and b, and CqPYL8a and b. On the contrary, the remaining CqPYLs showed some degree of variability in the UTR–exon–intron architecture, suggesting an active evolution process of these genes.

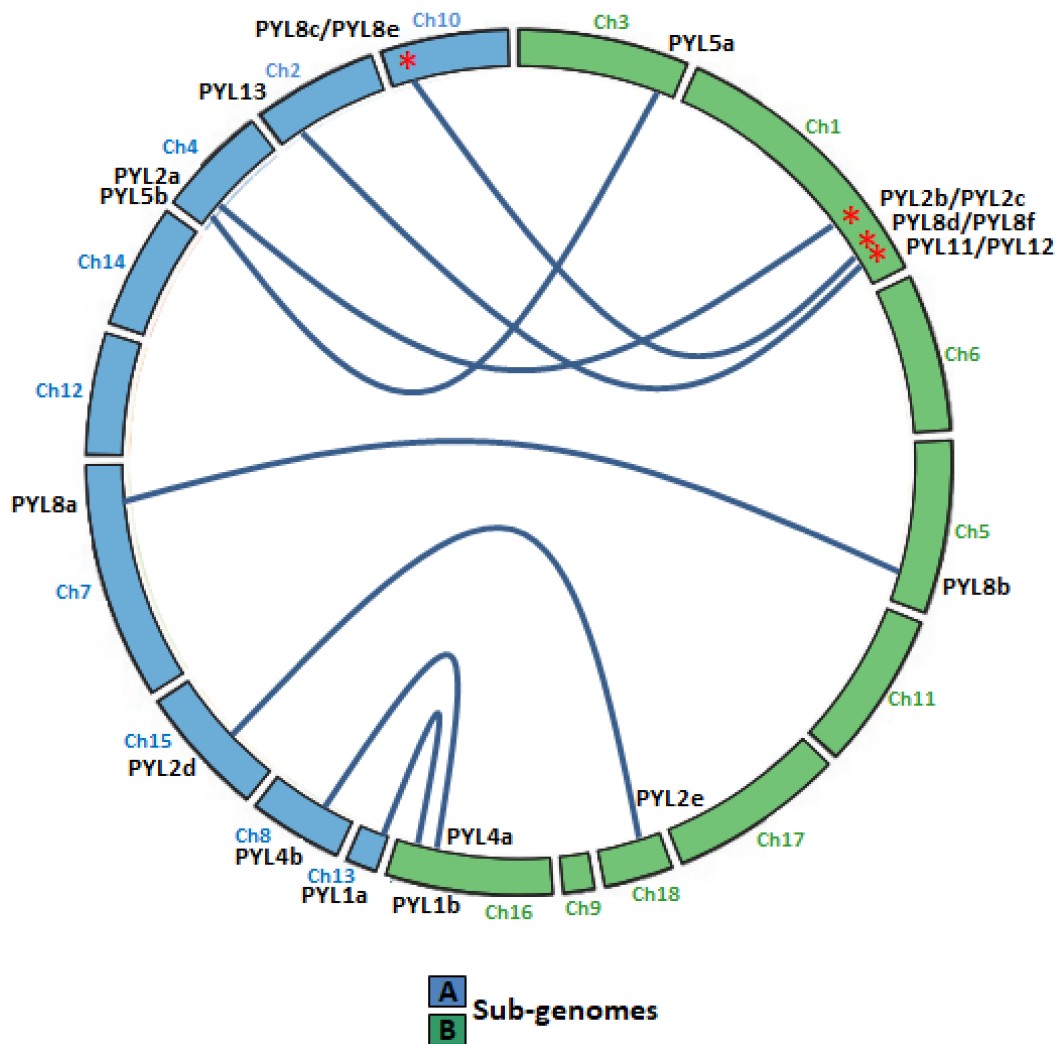

**Figure 3.** Collinear relationship between CqPYL genes. Homoeologous gene pairs in the A (blue chromosomes) and B (green chromosomes) sub-genomes are connected by blue lines. Red asterisks: paralogous CqPYL genes, proposed from a putative chromosome segmental duplication or pair of tandem duplication genes. Ch: chromosome number.

### 3.3. Expression Profiles of CqPYLs

The expression profiles of CqPYLs were investigated using public RNA-seq data (Figure 5). Results showed a differential expression of the 20 CqPYLs in 1-week-old seedlings (Figure 5a). The CqPYLs with relatively higher expression levels were: CqPYL8c-d and CqPYL4a-b. Followed by CqPYL8a-b, CqPYL5a-b, CqPYL8e-f and CqPYL1a-b. The remaining CqPYLs do not show significant expression levels in seedlings. Furthermore, CqPYLs displayed strong tissue specificity. In general, subfamily I of CqPYLs showed the highest expression levels with respect to subfamily II and III (Figure 5b). In particular, CqPYL8c-d showed preferential expression in the shoot from seedlings as well as adult leaves and stems. By contrast, CqPYL8a-b and e-f were highly expressed in roots. Furthermore, the members of subfamily II CqPYL4a-b showed a higher expression level in seedling roots, as well as whole seedlings and young seedlings, compared to other tissues (Figure 5b). On the other hand, the members of subfamily III had very low expressions in all tissues analyzed, with the exception of CqPYL1a-b, which showed some degree of expression in young seedlings and root seedlings (Figure 5b).

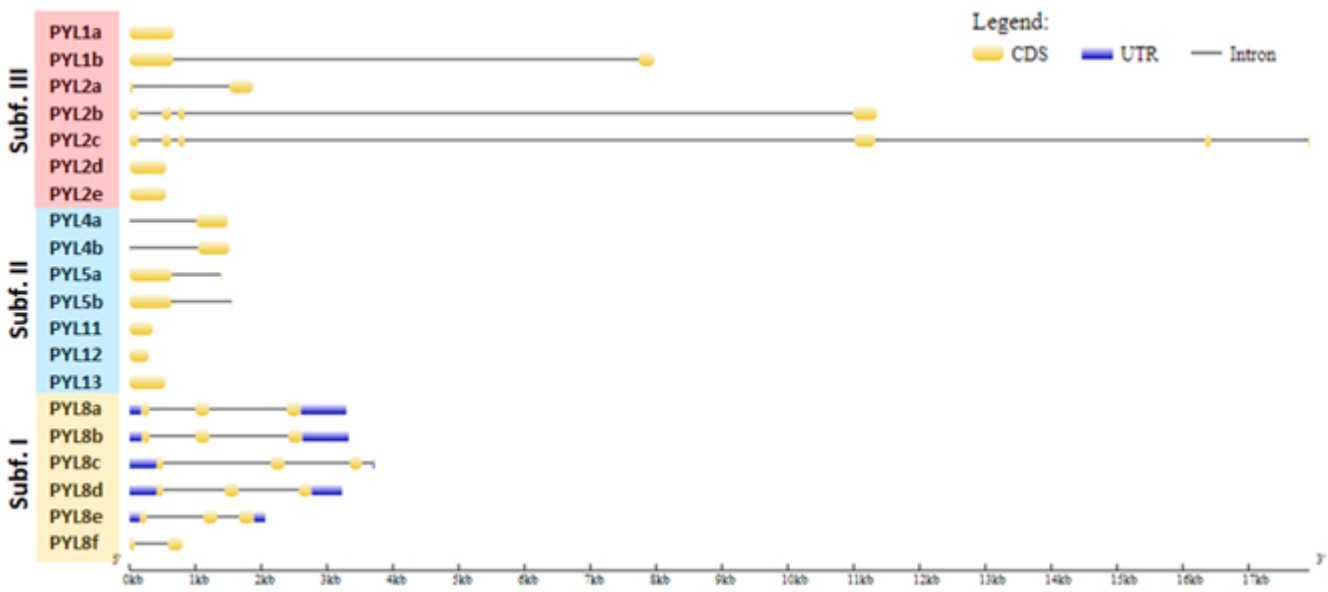

**Figure 4.** Gene structure of PYL genes in quinoa. The yellow and blue rectangles represent the CDSs and UTRs, respectively, and the black lines represents introns. The lengths of the CDSs, UTRs and introns for each PYL gene are shown proportionally. Orange, blue and red clades represent subfamilies I, II and III, respectively.

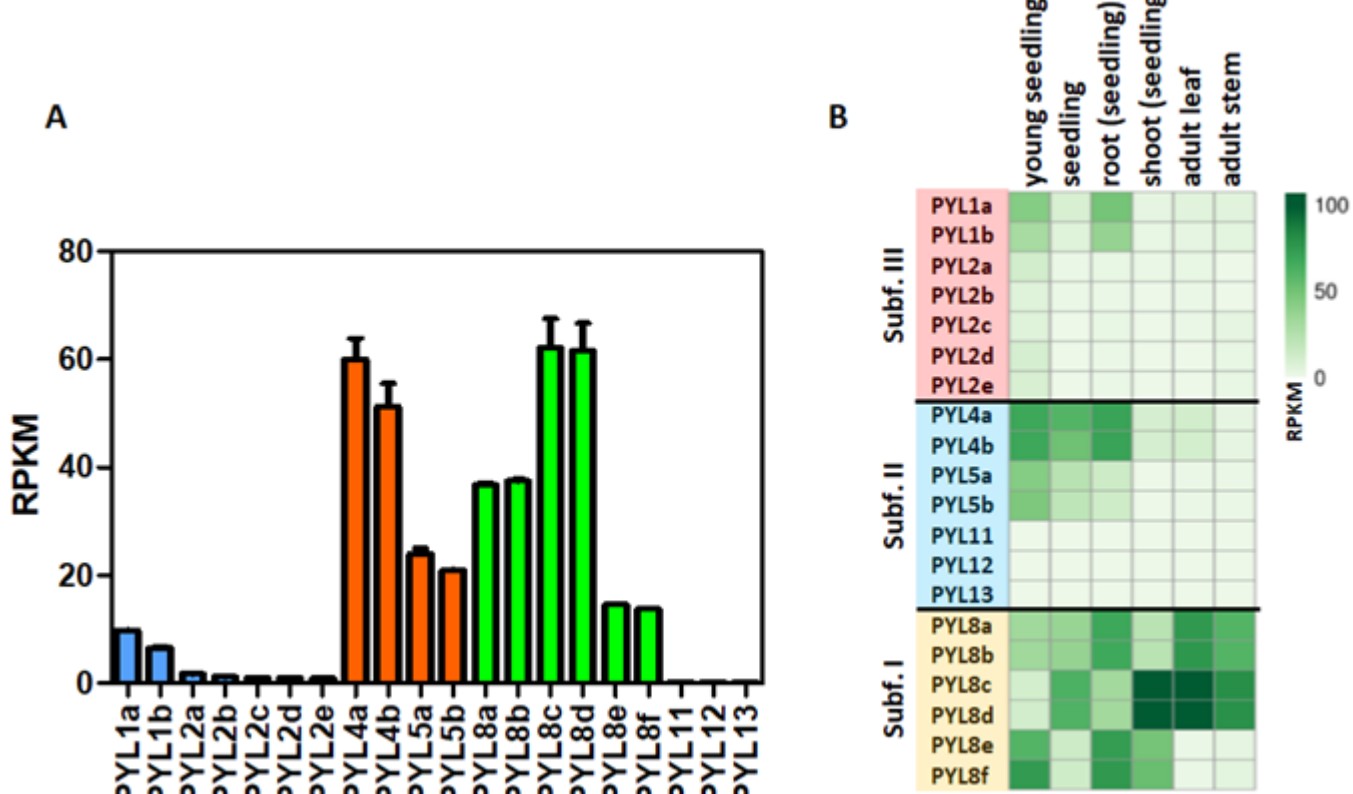

**Figure 5.** Expression patterns of CqPYLs. (**A**) Expression of CqPYLs in seedlings. (**B**) Heat-map of CqPYLs expression data from different tissues. RNA-seq samples from SRA-NCBI of different tissues were analyzed. The RPKM (reads per kilo base of transcript per million fragments mapped reads) value of each CqPYL was calculated and plotted.

### 3.4. Conserved Motif and Protein Alignments Analysis of PYLs in Quinoa

The presence of the PYR_PYL_RCAR motif domain in each PYL protein sequence was evaluated using the MOTIF search tool [44]. The MOTIF search was performed in the pfam and NCBI-CDD databases. Of note, the PYR_PYL_RCAR motif includes the Polyketide cyclase/dehydrase and lipid transport motif and the Bet_v1-like motif of major pollen allergen of *Betula verrucosa*. The 20 CqPYL identified in this work showed the presence of the PYR_PYL_RCAR motif domain (Figure 6). A close inspection of the amino acid sequence alignment of the 14 AtPYLs and the 20 CqPYLs shows a lack of the gate loop [45] in CqPYL11 and CqPYL12 (Figure S2). Moreover, CqPYL11 also shows an abnormally long latch loop. On the other hand, CqPYL8f lacks the N-terminal domain and the initial methionine (Figure S2). As expected, sub-genome B has had independent evolution from A and possibly experienced some degree of degradation or plasticity, despite the fact that it was proposed a lower loss or remodeling of the quinoa subgenome B with respect to A [29,46].

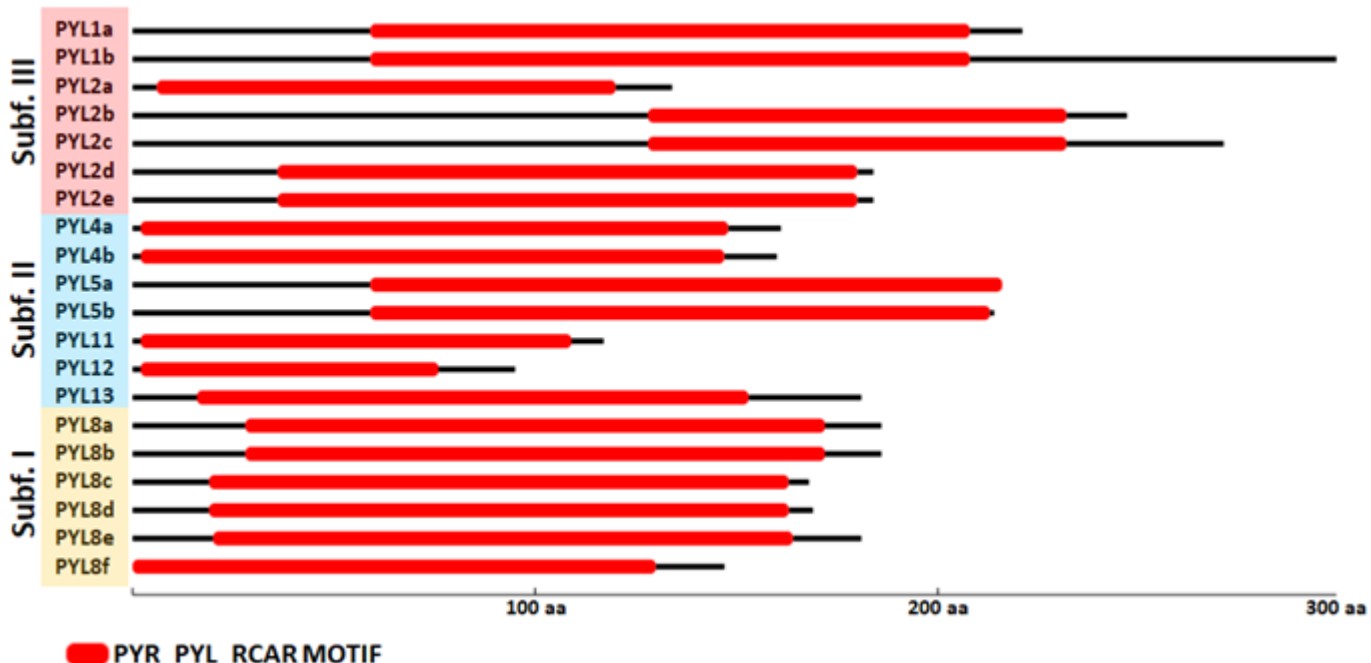

**Figure 6.** Conserved motifs analysis of CqPYL genes. The conserved motifs of the CqPYLs were identified using the MOTIF search tool. The MOTIF search was performed in pfam and NCBI-CDD databases. In red rectangles the PYR_PYL_RCAR motif domain (PYR_PYL-RCAR motif includes the Polyketide cyclase and the Bet_v1-like motif).

For a more detailed study of CqPYL proteins, alignments were performed for each ABA receptor subfamily. Then, alignments of some AtPYLs and CqPYLs, and the secondary alignment of ABA receptors were generated using ESPRIPT [36]. The secondary structure of the NbPYLs was inferred from the crystallographic structure of AtPYL1 (Protein DataBank Code 3kdj), AtPYL5 (4jdl), or AtPYL9 (3x9r) as appropriate. Alignments of AtPYLs and CqPYLs from subfamily I and III showed a high degree of conservation in both gate and latch loops, as well as in the protein residues involved in ABA coordination (Figure S3 and Figure S4, respectively). This fact suggests these receptors are likely functional. Although, CqPYL1b showed a C-terminal domain exceptionally long (more than 50 residues). On the contrary, alignments of AtPYL and CqPYL members of subfamily II showed some differences in CqPYL5a and CqPYL13 with respect to the Arabidopsis counterparts (Figure 7).

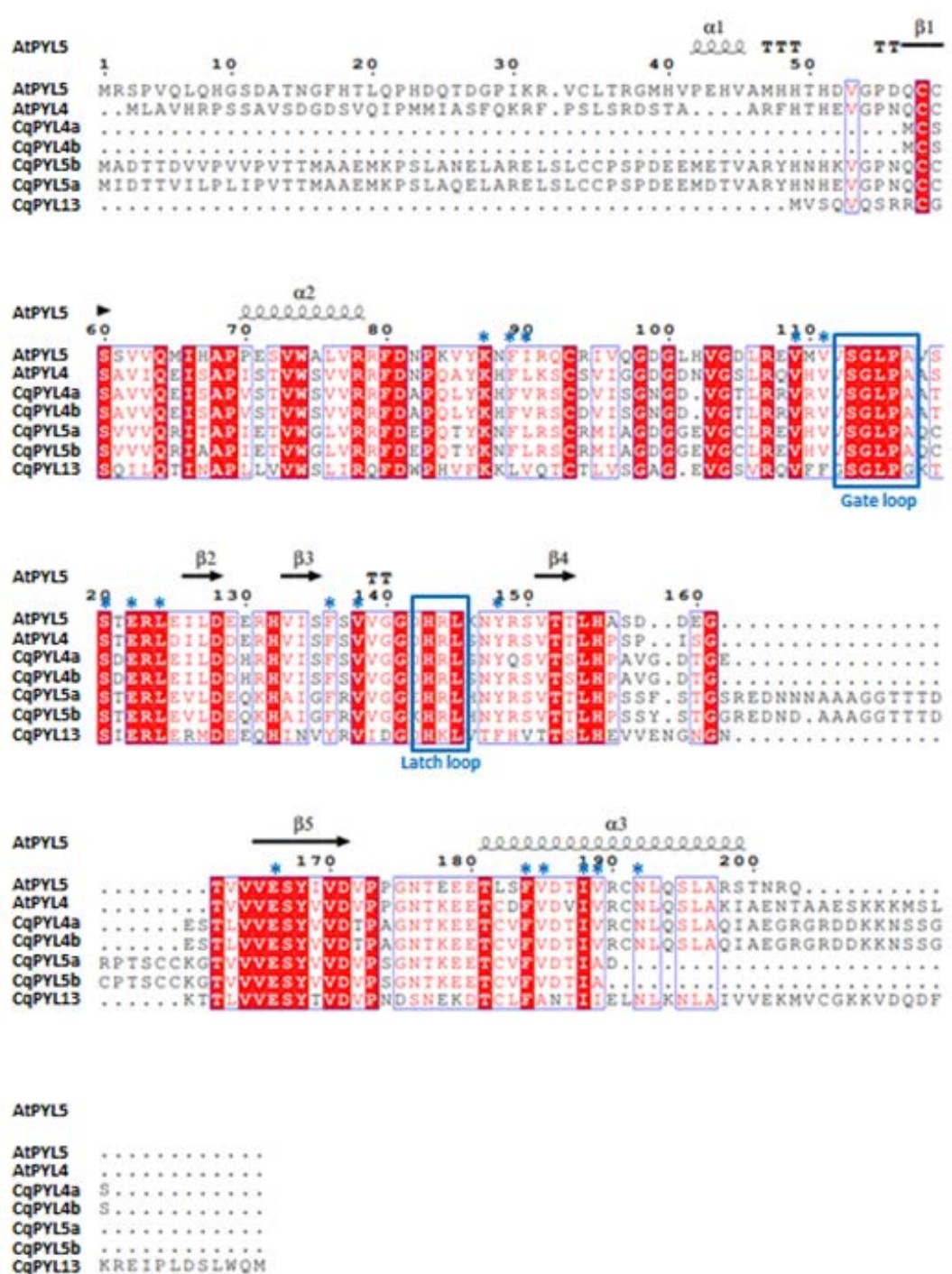

**Figure 7.** Amino acid sequence alignment of At and Nb sub-family II members of ABA receptors. Sequence and secondary structure alignment of ABA receptors are indicated. The secondary structure of the NbPYLs is predicted according to the crystallographic structure of AtPYL5 (Protein DataBank Code 4jdl), and was generated using the ESPRIPT tool [36]. Blue boxes indicate the position of the gate and latch loops. Blue asterisks mark residues involved in interactions with ABA.

Even though gate and latch loops are well conserved, CqPYL13 showed an "HKL" motif instead of the classical "HRL" motif for the latch loop. Probably, this change of arginine–lysine (R–K) is not interfering with the protein function by itself, but lysine could be a target of ubiquitination and/or sumoylation. Moreover, CqPYL13 also has variation in four residues (L41, F62, Y87, and F99) involved in the coordination with ABA, with the potential to change ligand affinity (Figure 7). Secondly, CqPYL13, together with

CqPYL4a-b, showed a shortened (around 50 residues) N-terminal domain. On the other hand, CqPYL5a-b showed a short C-terminal domain and also showed the presence of an unreported big loop between β4 and β5 sheets (Figure 7). This TTT loop includes a putative phosphorylation HOT-SPOT [47] with a tandem of three threonines (T176-178), followed by an extra threonine at three residues downstream (T182). In agreement, phosphorylation prediction on CqPYL5a with NetPhos 3.1 in silico tool showed a positive value in these four threonines (T176: 0.573, T177: 0.676, T178:0.983 and T182: 0.677). The NetPhos prediction score is a value in the range of 0.000–1.000; the scores above 0.500 indicate a phosphorylation target positive predictions. On the other hand, regarding the C-terminal domain of CqPYL5a-b, it was noticed the loss of a residue involved in ABA coordination, and possibly these receptors could have less affinity for ABA than other receptors of the same subfamily II. Moreover, the lack of half of the α3 helix could also disturb the interaction with the PP2C phosphatases.

### 3.5. Protein Homology Modeling and 3D Structure Analysis of CqPYLs

To further inspect the new features identified in CqPYL5a-b and CqPYL13, the 3D structure of some CqPYL proteins was obtained by homology modeling using the Swiss-Model in silico tool [37]. The models of CqPYL1a, CqPYL4a, and CqPYL13 were obtained using the topology of SlPYL1 (5mob) as a template; for CqPYL8a and CqPYL5a, protein 3D structure templates AtPYR1 (3k3k) and CsPYL1 (6zuc) were used, respectively (Figure 8 and Figure S5).

Ubiquitination regulates protein stability, localization, activation, and interactions [48,49]. The latch loop of CqPYL13 showed an "HKL" motif instead of the classical "HRL" motif. The structural alignment of CqPYL13 and AtPYL5 suggests that probably this change of arginine–lysine (R–K) is not interfering with the normal latch loop conformation in steric terms. By contrast, lysine could be a target of ubiquitination and/or sumoylation, and in that case, the dynamics of interaction with ABA and/or the formation of the ternary complex with PP2Cs could be greatly affected. As we can see in the CqPYL13 surface model, K95 is perfectly accessible to putative ubiquitination and/or sumoylation enzymes (Figure 9a). On the other hand, regarding the polymorphism in residues of CqPYL13 involved in the coordination with ABA, it is of particular interest that CqPYL13-F62 is bigger than the equivalent AtPYL1-V110 and could be modifying the ABA pocket size and concomitantly the ABA affinity (Figure 9b). AtPYL1-V110 is in coordination with ABA through a non-polar contact.

Protein phosphorylation is a fundamental mechanism through which protein function is regulated in response to extracellular stimuli [50]. The new TTT loop identified in CqPYL5a-b between β4 and β5 sheets it is relatively big with respect to other PYLs (Figure 10a). This TTT loop includes a putative phosphorylation HOT-SPOT, with four phosphorylatable threonines located on the protein surface (Figure 10a) and therefore accessible to kinases. In regards to the short C-terminal domain of CqPYL5a-b, it was noticed a loss of a residue involved in ABA coordination (Figure 10b). As we can see in the structural alignment of CqPYL5a and AtPYL1, the residue AtPYL1-N195 is in coordination with ABA, contributing to the affinity of the ternary complex ABA-PPYL-PP2C formation (Figure 10b). Moreover, the partial loss of the α3 helix could also disturb the interaction with the PP2C phosphatases (Figure 10c).

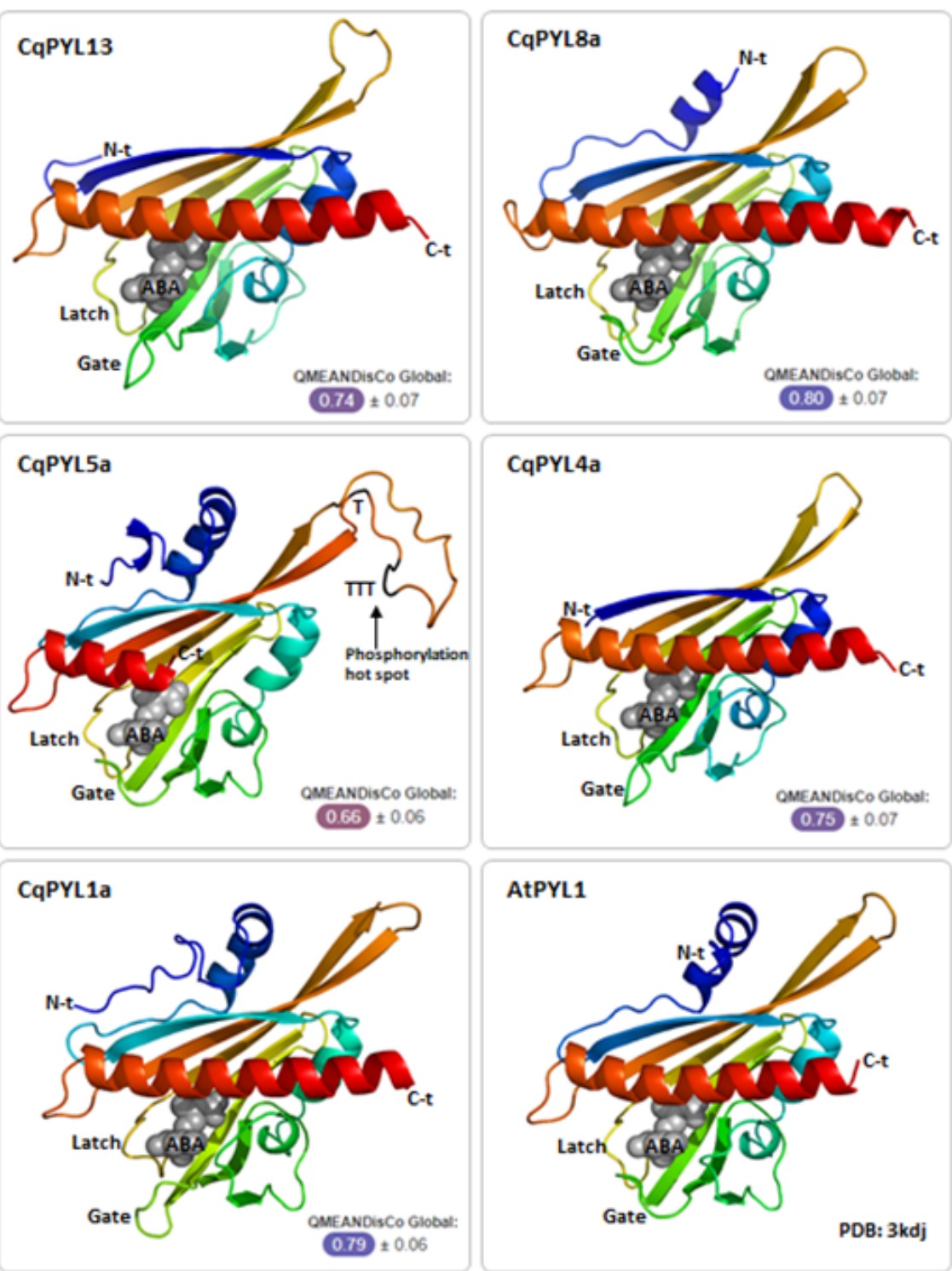

**Figure 8.** CqPYL 3D protein structure. CqPYLs topology were predicted by homology modeling using Swiss-Model and showed as cartoons using PyMOL. A quality estimation by QMEANDisCo global score was provided for each model. Gate and latch loops were indicated. ABA: abscisic acid; N-t: N-terminal domain; C-t: C-terminal domain.

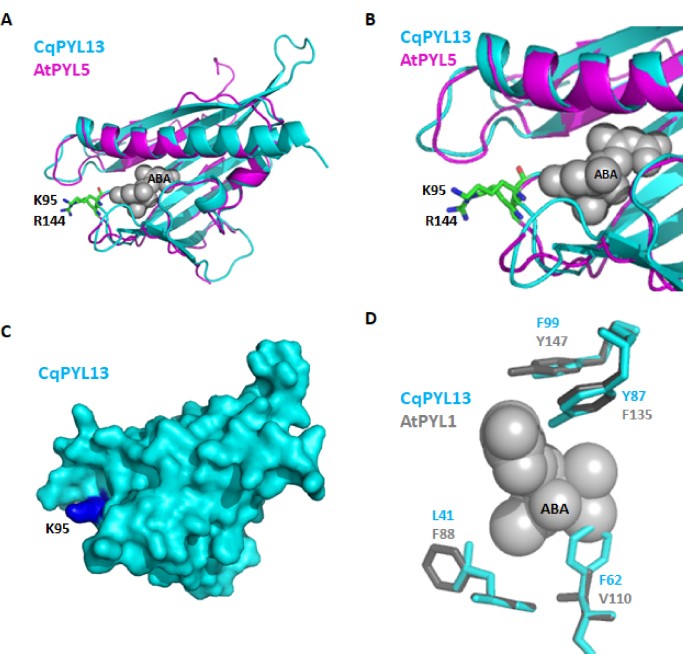

**Figure 9.** CqPYL13 topology analysis. (**A**). Structural alignment of CqPYL13 and AtPYL5. Protein structures were showed as cartoons. Residue K95 from CqPYL13 and R144 from AtPYL5 were indicated. (**B**) close-up from panel A. (**C**) CqPYL13 topology showed as surface. (**D**) Structural alignment of CqPYL13 and AtPYL1. Some protein residues from the ABA pocket were indicated and showed as stick view.

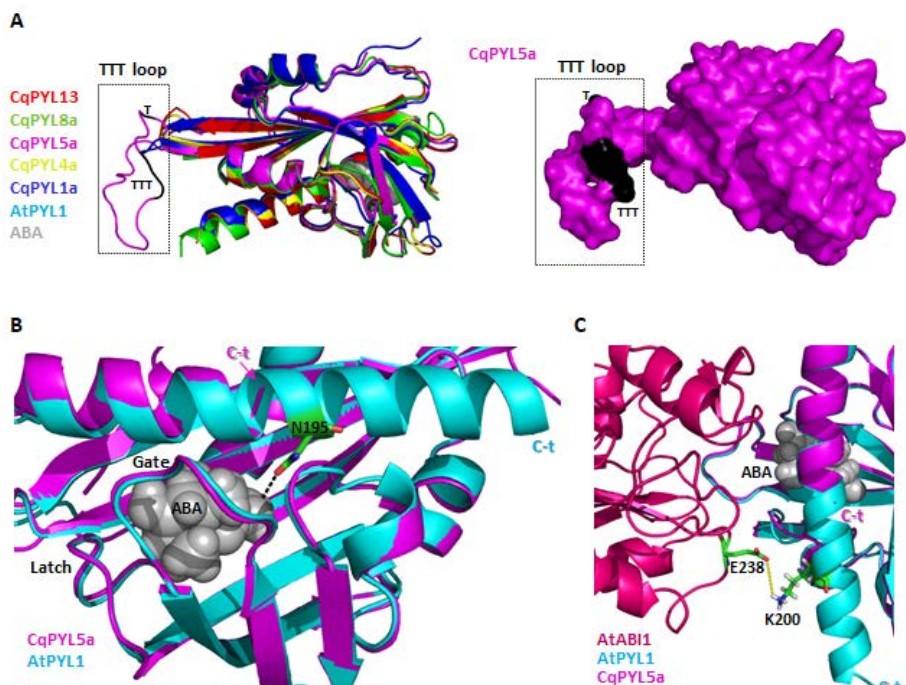

**Figure 10.** CqPYL5 topology analysis. (**A**). Structural alignment of CqPYL8a, CqPYL5a, CqPYL4a, CqPYL1a, AtPYL1 showed as cartoons (left) and CqPYL5a protein suface (right). TTT loop was indicated. (**B**). Structural alignment of CqPYL5a and AtPYL1 showed as cartoon. Gate and latch loop were indicated. Residue N95 from AtPYL1 was showed as stick view. C-t: C-terminal domain. (**C**) Protein structure of the arabidopsis ternary complex PYL1-ABA-ABI was aligned to CqPYL5a and showed as cartoon. Residues E238 and K200 from AtABI1 and AtPYL1, respectively were showed as sticks. Yellow dotted line: a putative polar coordination.

## 4. Discussion

Abscisic acid (ABA) is a key plant regulator of several agronomical traits, such as seed maturation and germination, shoot and root growth, as well as physiological responses to abiotic and biotic stresses [11–13]. The PYL gene family of ABA receptors is the initial step of the ABA signal transduction pathway. Identification and functional validation of these PYLs in ABA perception hold the promise to breed improved agronomic traits in crops (e.g., stress tolerances) to cope with the challenge of global climate change [16,41,51–55].

In the last years, the PYL gene family has been identified in several crops, including *N. benthamiana* [28], *P. dactylifera* [56], *S. lycopersicum* [22], *O. sativa* [23], *G. hirsutum* [57], *T. aestivum* [27], *V. Vinifera* [26] *and Z. mays* [58] between others. Based on the genome-wide search method, we identified 20 PYL genes in *C. quinoa*. These PYLs could be divided into three subfamilies, accordingly to previous studies in Arabidopsis and other species: subfamily I or PYL8 subgroup, subfamily II with PYL4 subgroup, and subfamily III or PYL1 subgroup [13,28,35–37]. Quinoa is a resilient allotetraploid crop, well adapted to growth in sub-optimal environments. The evolutionary history of angiosperms showed polyploidization in many species [59,60]. Moreover, polyploidy contributes to gene sub-functionalization and diversification, leading to plant evolution and adaptation [60–62]. For instance, polyploidy contributed to the colonization of harsh geographical niches, conferring abiotic stress tolerance in different plant species [61,63]. The selection pressure wielded by the harsh conditions of quinoa growth niches steered to strict control of the plant water use efficiency, probably facilitated by a double set of ABA receptors in the genome.

All quinoa PYL genes had the A and B homoeologous copies in the quinoa A and B subgenome. Hence, it was proposed eight homoeologous sub-groups in quinoa PYL genes (Figure 3 and Figure S1): CqPYL1a-b, CqPYL2a-c, CqPYL2d-e, CqPYL4a-b, CqPYL5a-b, CqPYL8a-b, CqPYL8c-f, and CqPYL11-13. Moreover, genomic segmental duplication or gene tandem duplication was observed in some CqPYLs. At least four duplications events occurred in quinoa evolution, giving rise to the paralogous genes CqPYL8c/e, CqPYL11/12, CqPYL8d/f, and CqPYL2b/c (Figures 1 and 3). On the other hand, the UTR–exon–intron organization of CqPYL reveals some degree of variability in the gene architecture, suggesting an active evolution process of these genes. Although, it was proposed that the subgenomes of quinoa are exceptionally well conserved by structure and sequence [45]. Nevertheless, found here are some aberrant CqPYLs (CqPYL11, CqPY12, and CqPYL8f). Further studies of the specific patterns of PYL gene retention or dispersion might provide clues to understanding the chromosome interaction and genetic evolution during quinoa allopolyploidization.

Many studies have found that PYL genes were expressed in different tissues, including seeds, roots, leaves, flowers, and fruits [22]. The CqPYL8c-d and CqPYL4a-b showed the higher expression level, followed by CqPYL8a-b, CqPYL5a-b, CqPYL8e-f, and CqPYL1a-b. Moreover, each pair of homoeologous genes showed no significant differential expression, suggesting an absence of their sub-functionalization. Although this observation is not conclusive since homoeologous CqPYL receptors showed high nucleotide sequence identity and several RNA-seq reads in the alignment were ambiguous, therefore we could not unequivocally assign them. On the other hand, CqPYLs displayed strong tissue specificity. CqPYL8c-d has higher expression levels in seedling shoots and adult leaves, while CqPyl8a-b, CqPYL4a-b, and CqPYL1a-b were predominantly expressed in roots. *AtPYL8* has an important role in regulating lateral root growth in the presence of ABA [64]. Nevertheless, in quinoa, members of the three sub-families have the potential to be regulators of root growth in response to the ABA signal. In general, subfamily I of CqPYLs showed the highest expression levels with respect to subfamily II and III (Figure 5b). This differential expression pattern in PYLs sub-families is something expected in crops, especially in species well adapted to harsh conditions, such as date palm and *N. benthamiana*. It was reported in *P. dactylifera* that four members of subfamily I (PYL8-like) and two members of sub-family III (PYL1-like) were preferentially expressed, whereas gene expression of sub-family II members was low or almost undetectable [56]. Moreover, in *N. benthamiana* two genes of

subfamily III and one gene of subfamily I showed the highest expression among all ABA receptors, and NbPYL members of subfamily II showed the lower expression [28].

Quinoa exhibits high tolerance to certain abiotic stresses, particularly salinity and drought [65]. These traits are related to ABA signaling and stomatal closure [12,66]. In contrast, quinoa is not very well adapted to high temperatures environments [65,67]. Hot climates represent a serious challenge in the cultivation of quinoa, causing the abortion of flowers and the death of pollen [67]. Interestingly, the plant tolerance to high temperatures requires stomatal opening to induce leaf cooling [68,69]. Could the evolutionary modification of the CqPYL genes possibly be molecular evidence for the partial loss of heat tolerance at the expense of improving tolerance to salinity and drought? Even more, drought and heat stress often coincide in nature [70] and may trigger conflicting responses, as the individual signals trigger opposite stomatal responses. A comparison of CqPYL sequences and expression patterns of these genes across an "adaptive spectrum" of different genotypes (i.e., a highly heat-tolerant *C. hircinum* from northern Argentina, a moderately heat-tolerant QQ74 from Chile, and heat-sensitive Andean quinoa like Real) could shed light on these issues.

The 20 CqPYLs showed the presence of the PYR_PYL_RCAR motif domain, suggesting that all of them have the potential to be functional ABA receptors (Figure 6). Nevertheless, the lack of the gate loop in CqPYL11 and CqPYL12 and the abnormally long latch loop observed in CqPYL11 indicate that these receptors, in particular, could be non-functional. The gate and latch loop are key futures involved in the PYLs function [45]. Interestingly CqPYL11/12 are paralogous genes from subgenome B. Moreover, CqPYL8f is located in subgenome B and lacks both the N-terminal domain and the initial methionine (Figure S2). As expected, sub-genome B has had independent evolution from A and possibly experienced some degree of degradation or plasticity, despite the fact that it was proposed a lower loss or remodeling of the quinoa subgenome B with respect to A [29,46]. On the other hand, gate and latch loops are well conserved in CqPYL13, but this receptor showed an "HKL" motif instead of the classical "HRL" motif for the latch loop. This lysine (K) could be a target of ubiquitination and/or sumoylation, which would include a new regulation mechanism in ABA receptors. CqPYL13 also showed some changes in the ABA pocket with the potential to modify ABA and agonists coordination. Otherwise, CqPYL5a-b showed a TTT loop with a phosphorylation HOT-SPOT. Of note: A phosphorylation HOT-SPOT is defined as one containing four phosphorylatable residues within 10 consecutive amino acids [47]. In agreement, a positive phosphorylation prediction on CqPYL5a was observed in this TTT loop using NetPhos. Moreover, the lack of the AtPYL1-N195 equivalent residue in CqPYL5a-b could diminish its affinity for ABA. Furthermore, the lack of a complete α3 helix could also interfere with the PP2C phosphatases interaction. For instance, the residue AtPYL1-K200 is near (less than 5 Angstrom) AtABI1-E238, possibly establishing polar coordination and contributing to the ternary complex stabilization. The lack of AtPYL1-K200 equivalent residue in CqPYL5a-b could be diminishing the affinity with CqPP2c phosphatases.

In conclusion, the 20 CqPYLs identified in this work provide the fundamental basis for the characterization of ABA signaling in quinoa and for further stress-resistance breeding in crops.

**Supplementary Materials:** The following supporting information can be downloaded at: https://www.mdpi.com/article/10.3390/stresses2030021/s1. Figure S1: Genetic distance matrix of CqPYLs CDS sequences; Figure S2: Amino acid sequence alignment of the 14 AtPYLs and the 20 CqPYLs; Figure S3: Amino acid sequence alignment of At and Nb sub-family III members of ABA receptors; Figure S4: Amino acid sequence alignment of At and Nb sub-family I members of ABA receptors; Figure S5: Swiss-Model CqPYL protein homology modeling.

**Funding:** This research received no external funding.

**Conflicts of Interest:** The authors declare no conflict of interest.

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
