# Peer review of "Genome-Wide Identification of the PYL Gene Family in Chenopodium quinoa: From Genes to Protein 3D Structure Analysis"

_stresses, doi:10.3390/stresses2030021_

Round 1
Reviewer 1 Report
Line 179-180, p 5: Kolano et al. (ref. 42) did not show that C. pallidicaule was the actual AA ancestor; in fact, their GISH data indicated C. nevadense and C. neomexicanum were much closer than C. pallidicaule. However, the comparison with C. pallidicaule in the Figure 3 Circus Plot is informative since it does belong to the AA-diploid group.
Contextualization/discussion of results: Some mention, at least, of the quinoa genotype (QQ74, in this case) is absolutely necessary. QQ74 is a moderately heat-tolerant genotype, but cultivated quinoa in general is notoriously heat-sensitive (as is the compared AA diploid, C. pallidicaule). Could the evolutionary modification of the CqPYL genes possibly be molecular evidence for the partial loss of heat tolerance? Your argument seems to be framed within the context that quinoa has exceptional abiotic stress tolerance, but that is only partially true. It would be very interesting to compare sequences and expression patterns of these genes across an "adaptive spectrum" of AABB genotypes: a highly heat-tolerant C. hircinum from northern Argentina; moderately heat-tolerant QQ74; and a heat-sensitive Andean quinoa like Real. I think you absolutely need to inject some sort of discussion like this into your manuscript.
Author Response
Line 179-180, p 5: Kolano et al. (ref. 42) did not show that C. pallidicaule was the actual AA ancestor; in fact, their GISH data indicated C. nevadense and C. neomexicanum were much closer than C. pallidicaule. However, the comparison with C. pallidicaule in the Figure 3 Circus Plot is informative since it does belong to the AA-diploid group.
Sure, you are right, my mistake. Corrections were made, and this is the new paragraph: “Quinoa resulted from the hybridization of the ancestral diploid genomes A and B [42]. Accordingly to the genomic in situ hybridization results, possibly from C. nevadense or C. watsonii as donor of genome A and from C. suecicum, C. ficifolium or C. viridediploid as donor of genome B [42].
Contextualization/discussion of results: Some mention, at least, of the quinoa genotype (QQ74, in this case) is absolutely necessary. QQ74 is a moderately heat-tolerant genotype, but cultivated quinoa in general is notoriously heat-sensitive (as is the compared AA diploid, C. pallidicaule). Could the evolutionary modification of the CqPYL genes possibly be molecular evidence for the partial loss of heat tolerance? Your argument seems to be framed within the context that quinoa has exceptional abiotic stress tolerance, but that is only partially true. It would be very interesting to compare sequences and expression patterns of these genes across an "adaptive spectrum" of AABB genotypes: a highly heat-tolerant C. hircinum from northern Argentina; moderately heat-tolerant QQ74; and a heat-sensitive Andean quinoa like Real. I think you absolutely need to inject some sort of discussion like this into your manuscript.
Really interesting topic. This paragraph was added to discussion and some more citations: Quinoa exhibits high tolerance to certain abiotic stresses, particularly salinity and drought [66]. These traits are related with ABA signaling and the stomatal closure [12,67]. In contrast, quinoa is not very well adapted to high temperatures environments [66,68]. Hot climates represent a serious challenge in the cultivation of quinoa, causing abortion of flowers and death of pollen [68]. Interestingly, the plant tolerance to high-temperatures requires stomatal opening to induce leaf cooling [69-70]. Could the evolutionary modification of the CqPYL genes possibly be molecular evidence for the partial loss of heat tolerance, at the expense of improving tolerance to salinity and drought? Even more, drought and heat stress often coincide in nature [71] and may trigger conflicting responses, as the individual signals trigger opposite stomatal responses. A comparison of CqPYL sequences and expression patterns of these genes across an "adaptive spectrum" of different genotypes (i.e. a highly heat-tolerant C. hircinum from northern Argentina, a moderately heat-tolerant QQ74 from Chile, and a heat-sensitive Andean quinoa like Real) could shed light on these issues.
Reviewer 2 Report
The manuscript ‘Stresses-1832585’ deals with very interesting and important topic of new family genes PYL in Chenopodium quinoa using in situ analysis. The presented manuscript is reasonably balanced and well written. Therefore, it can be conditionally accepted with a subject to minor but very important revision. The author has to address comments and suggestions indicated below.
Minor comments/corrections:
(1) Starting from the Title and through each part of the manuscript, the author has to insert the term ‘in situ’ indicating that the experiments were made by other researchers but the author used results from publicly available databases. Please insert ‘in situ’ in the Title and as well as in Introduction, M&M, and Results sections.
(2) L50. English has to be improved in the phrase: “…drought resistance tolerance…”. This is better to use ‘drought tolerance’ only.
(3) L63. English has to be improved in the phrase: “ABA response plays a key role in plant response to abiotic…”. The term ‘response’ is used two times nearby.
(4) L71 and in other parts. The name ‘benthamiana’ is absurd because it has to be botanical name ‘Nicotiana benthamiana’ or ‘N. benthamiana’, as it is mentioned in L89. Please correct.
(5) L89-90. Please provide here the sequence ID numbers for each of the used PYL protein from Arabidopsis thaliana and Nicotiana benthamiana.
(6) L109 and in other parts. Please use ‘UTR’ in Uppercase because this is the abbreviation for ‘Untranslated region’.
(7) L118. Please correct the botanical name Betula verrucosa in Italics and please explain why white birch was used for major pollen allergen analysis?
(8) L121. The author’ statement about plant stage as ‘15 hours old young seedlings’ is unclear. Please re-phrase, for example as follows ‘young seedlings after 15 hrs of [what?]’. However, if 15 hrs were applied after seeds moisturising or germination, these are not seedlings but germinated seeds.
(9) L355-356. Please check and correct that all Latin names have to be in Italics.
(10) L437. Please consider to add one conclusive sentence or paragraph in the end of the manuscript.
Author Response
The manuscript ‘Stresses-1832585’ deals with very interesting and important topic of new family genes PYL in Chenopodium quinoa using in situ analysis. The presented manuscript is reasonably balanced and well written. Therefore, it can be conditionally accepted with a subject to minor but very important revision. The author has to address comments and suggestions indicated below.
Minor comments/corrections:
(1) Starting from the Title and through each part of the manuscript, the author has to insert the term ‘in situ’ indicating that the experiments were made by other researchers but the author used results from publicly available databases. Please insert ‘in situ’ in the Title and as well as in Introduction, M&M, and Results sections.
Sorry. Do you mean “in-silico”? If yes, sure I can add this term in title and other sections. I think that "in-situ" is not the right term to define this work. On the other hand, the only experimental data used in this work came from public RNAseq and this is explained along the manuscript.
(2) L50. English has to be improved in the phrase: “…drought resistance tolerance…”. This is better to use ‘drought tolerance’ only.
Yes sure, this sentence was corrected accordingly.
(3) L63. English has to be improved in the phrase: “ABA response plays a key role in plant response to abiotic…”. The term ‘response’ is used two times nearby.
That is right, this sentence was improved.
(4) L71 and in other parts. The name ‘benthamiana’ is absurd because it has to be botanical name ‘Nicotiana benthamiana’ or ‘N. benthamiana’, as it is mentioned in L89. Please correct.
Corrected.
(5) L89-90. Please provide here the sequence ID numbers for each of the used PYL protein from Arabidopsis thaliana and Nicotiana benthamiana.
Sequence ID was provided.
(6) L109 and in other parts. Please use ‘UTR’ in Uppercase because this is the abbreviation for ‘Untranslated region’.
UTR (in uppercase) were corrected.
(7) L118. Please correct the botanical name Betula verrucosa in Italics and please explain why white birch was used for major pollen allergen analysis?
Corrections were made and “White birch” was removed (it was wrong).
(8) L121. The author’ statement about plant stage as ‘15 hours old young seedlings’ is unclear. Please re-phrase, for example as follows ‘young seedlings after 15 hrs of [what?]’. However, if 15 hrs were applied after seeds moisturising or germination, these are not seedlings but germinated seeds.
This definition came from the SRA (SRR10493379) RNAseq data sheet. An explanation (sterilized seeds were embedded on solid 1/2 MS medium for 15h) was added in methods. Although during germination we only see the emergence of the root tip, the seedling is inside the seed, even before the start of germination.
(9) L355-356. Please check and correct that all Latin names have to be in Italics.
Corrected
(10) L437. Please consider to add one conclusive sentence or paragraph in the end of the manuscript.
Totally right, this sentence was added at the end: In conclusion, the 20 CqPYLs identified in this work, provides the fundamental basis for the characterization of ABA signaling in quinoa and for further stress-resistance breeding in crops.
Round 2
Reviewer 1 Report
The revised manuscript incorporated my editorial recommendations and is now suitable for publication.